# A Role for the Interactions between Polδ and PCNA Revealed by Analysis of *pol3-01* Yeast Mutants

**DOI:** 10.3390/genes14020391

**Published:** 2023-02-02

**Authors:** Shaked Nir Heyman, Mika Golan, Batia Liefshitz, Martin Kupiec

**Affiliations:** The Shmunis School of Biomedicine and Cancer Research, The George S. Wise Faculty of Life Sciences, Tel Aviv University, Tel Aviv 69978, Israel

**Keywords:** DNA polymerase, *Saccharomyces cerevisiae*, mutagenesis, homologous recombination, DNA damage tolerance, PCNA

## Abstract

Several DNA polymerases participate in DNA synthesis during genome replication and DNA repair. PCNA, a homotrimeric ring, acts as a processivity factor for DNA polymerases. PCNA also acts as a “landing pad” for proteins that interact with chromatin and DNA at the moving fork. The interaction between PCNA and polymerase delta (Polδ) is mediated by PIPs (PCNA-interacting peptides), in particular the one on Pol32, a regulatory subunit of Polδ. Here, we demonstrate that *pol3-01,* an exonuclease mutant of Polδ’s catalytic subunit, exhibits a weak interaction with Pol30 compared to the WT DNA polymerase. The weak interaction activates DNA bypass pathways, leading to increased mutagenesis and sister chromatid recombination. Strengthening *pol3-01′*s weak interaction with PCNA suppresses most of the phenotypes. Our results are consistent with a model in which Pol3-01 tends to detach from the chromatin, allowing an easier replacement of Polδ by the trans-lesion synthesis polymerase Zeta (Polz), thus leading to the increased mutagenic phenotype.

## 1. Introduction

DNA synthesis is an essential and universal process. Accurate DNA synthesis has an important role in preventing spontaneous mutations and carcinogenesis [1]. In the yeast *Saccharomyces cerevisiae,* DNA replication is carried out by three DNA polymerases that belong to the B family [2]. DNA synthesis is initiated by the activity of Polα. This polymerase is then replaced by either Polε or Polδ. Polε is currently believed to synthesize most of the leading strand [3], whereas Polδ synthesizes the lagging strand and creates Okazaki fragments [4].

The synthesis ability of Polδ depends on PCNA, a homotrimeric clamp (composed of three copies of the Pol30 protein) that encircles the DNA, enabling better DNA processivity during DNA synthesis [5]. The interaction between PCNA and Polδ is mediated by a conserved motif called the PCNA-interacting peptide (PIP). Its consensus sequence is Q-x-x(M/L/I)-x-x-F-F, although many variations of the PIP motif have been found [6,7].

Polδ is a complex composed of three subunits; Pol3 (the catalytic subunit), Pol31, and Pol32 [8,9] (Figure 1A). *POL3* and *POL31* are essential genes [10], while *POL32* is not [9], although its absence confers cold sensitivity and affects the repair of double-stranded breaks (DSBs) [11]. It has been shown that *POL32* has a PIP motif at its C terminus, through which it physically interacts with PCNA. Mutating the PIP motif of *POL32* or deleting the *POL32* gene altogether does not lead to lethality or to a significant decrease in the polymerase’s processivity [12]. This suggests that additional points of interaction between the polymerase subunits are likely to exist. A crystal structure of the holoenzyme revealed that Pol3’s interaction with Pol32 is mediated by Pol31 [13]. In addition, it was shown in vitro that mutation of potential PIP motifs in *POL3* or *POL31* causes a decrease in DNA synthesis processivity, as well as lethality in *pol32Δ* cells [14]. Together with experiments showing that Pol3 physically interacts with PCNA [15], this suggests that the interaction of Polδ with PCNA does not only rely on the binding of Pol32 to PCNA but also on the binding of Pol31 and/or Pol3 to PCNA, although the interaction of Pol32 with PCNA is the strongest.

During cell growth, DNA synthesis may be blocked by DNA lesions, covalently linked proteins, R-loops, secondary structures of the DNA, etc.; this situation can be lethal to the cell. However, the cell can bypass these impediments by using the DNA damage tolerance (DDT, also known as post-replication repair) pathways (reviewed in [16,17]). Two main modes of lesion tolerance have been described: (1) Mono-ubiquitination of PCNA at lysine 164 by the Rad6/Rad18 E2/E3 complex [18,19] activates an “error-prone pathway” that uses alternative, trans-lesion synthesis (TLS) polymerases that are able to extend the DNA synthesis by incorporating more or less random nucleotides facing any DNA lesion. This, of course, results in the creation of mutations but prevents cell arrest and death. In yeast cells, the main TLS polymerase is Polζ, encoded by the *REV3* and *REV7* genes [20]. The TLS pathway is responsible for 50–70% of all spontaneous mutations and for the increased frequency of mutations seen following DNA damage [21,22,23,24]. (2) In addition, the cells may opt to bypass the lesion through an ‘error-free pathway’(also called “template switch”) mechanism that uses the undamaged sister chromatid as a template in order to copy the correct sequence and allow the cell to bypass those lesions [25,26]. This bypass pathway involves Rad5, a ubiquitin ligase/helicase that, together with Ubc13 and Mms2, poly-ubiquitinates PCNA at lysine 164 [27,28].

Here, we test the importance of Polδ-PCNA interactions for the different DNA bypass pathways. The exonuclease domain in DNA polymerases carries out an important proofreading function, which allows the polymerase to detect and replace errors that may have occurred during polymerization. In humans, mutations in the exonuclease domain of DNA polymerases are common in various types of cancer [29,30]. We show that yeast strains with the *pol3-01* allele, which are mutated in the exonuclease domain of *POL3,* show elevated mutation and sister chromatid recombination phenotypes. *pol3-01′*s high mutation rate is usually attributed to its lack of exonuclease. In this paper, we test this assumption and show evidence that *pol3-01′*s high mutation rates are caused mainly by its weak interaction with PCNA.

## 2. Materials and Methods

### 2.1. Yeast Strains

The S. cerevisiae strains used in the present study (Table 1) are isogenic derivatives of BY4741 (MATa his3∆ leu2∆ met15∆ ura3∆), E134 (MAT@ ade5-1 lys2::InsEa14 trp1-289 his7-2 leu2-3,112 ura3-52) and PJ69-4a (MATa trp1-901 leu2-3,112 ura3-52 his3-200 gal4∆ gal80∆ GAL2-ADE2 LYS2:: GAL1-HIS3 met2::GAL7-lacZ). Gene deletions were performed by using a PCR-mediated one-step replacement technique. All deletions were confirmed by PCR amplification of genomic DNA and phenotypic expression. Site-directed mutagenesis was introduced by using a modified oligonucleotide in a PCR reaction, followed by transformation to yeast. All changes were confirmed by DNA sequencing. Cloning was performed by standard methods and was confirmed by restriction fragment analysis, PCR, and sequencing. The transformation was carried out by standard methods using lithium acetate.

### 2.2. Plasmids

Yeast two-hybrid plasmids were built by cloning the relevant fragments using *Sal*I and *Eco*RI for pGBKT7 and *Bam*HI and *Xho*I for pACT2. The YIpAM26 *pol3-01* plasmid was received from R. Kolodner [31]. The pACT2 *POL32* plasmid was received from P. Burgers. All plasmids are listed in Table 2.

### 2.3. Media and Growth Conditions

*Saccharomyces cerevisiae* strains were grown at 30 °C unless specified otherwise. The growth medium for all batch cultures was either standard minimal medium made of 6.7 g/liter yeast nitrogen base without amino acids and with ammonium sulfate, 1.5 g/liter amino acid dropout powder, and 2% of glucose: or standard YPD-rich medium containing 1% yeast extract, 2% bacto peptone, and 2% dextrose. Agar plates were made with the same growth medium plus 2% agar.

### 2.4. Yeast Two-Hybrid (Y2H) Assay

To detect two-hybrid interactions, yeast strain PJ69-4a was co-transformed with one *LEU2*-marked plasmid containing *POL30*/*POL32* genes fused to the *GAL4*-activating domain (pACT2) and one *TRP1*-marked plasmid containing *POL3*/*pol3-01*/*pol3-pip32* genes fused to the *GAL4* DNA binding domain (pGBKT7). Yeast cultures were grown in SD-Trp-Leu medium and spotted on SD-Trp-Leu plates and SD-Trp-Leu-His plates. Cells were incubated for 3–4 days at 30 °C.

### 2.5. Fluctuation Test

Derivatives of E134 were plated on media that allowed us to detect new mutations: CAN medium is a standard minimal medium that lacks arginine and contains canavanine, a toxic agent that is an analog to arginine. If loss of function mutations occur in the *CAN1* gene, the transporter loses its function, preventing the toxic canavanine from entering the cell and allowing the mutated cells to grow on this medium. The *LYS2* locus of E134-derived strains contains an insertion of 14 adenine residues, which causes auxotrophy to lysine. Frameshift mutations (+1 or −2) can restore the Lys+ phenotype. SD-Lys is a standard-defined medium that lacks lysine. The *trp1-289* mutation of E134-derived strains contains a single nucleotide change, which causes auxotrophy to tryptophan. Reversion to the original nucleotide allows the cells to grow on an SD-Trp plate, which is a standard-defined medium that lacks tryptophan.

Derivatives of BLS2 were plated on media that allowed us to detect new sister chromatid recombination: Strain BLS2 contains 5′ *ade3* and 3′ *ade3* fragments that are separated by *URA3* marker. This construct allows the detection of new unequal sister chromatid recombination (USCR) by plating the cells on the appropriate medium (SD-HIS) [32]. Mutation and USCR rates were calculated as described in [33].

### 2.6. Fractionation Analysis

Different E134 strains were grown to O.D. 1. After reaching the desired O.D., cells were treated with 100T zymolyase 20 mg/mL and 0.01 M/mL of DTT for 1 h. Cells were then washed with SB buffer (0.02M Tris and 1M Sorbitol) and lysed using EB buffer (0.02 M Tris and 0.1 M NaCl), and fractions were separated using NIB buffer (0.02 M Tris, 0.1 M NaCl, and 1.2 M sucrose), using a sucrose gradient. A total of 2 mg of protein were extracted for each fraction: whole cell extract (WCE) and chromatin.

### 2.7. Protein Extraction and Immunoprecipitation Assays

Cells were grown to mid-logarithmic phase, washed once with water, and resuspended in lysis buffer (PBS, pH 7.0, 200 mM NaCl, 0.5 mM EGTA, 0.5 mM EDTA, 0.1% Triton X-100, protease inhibitor mixture, and 1 mM phenylmethylsulfonyl fluoride). Cells were broken by bead beating (45 min at 4 °C) with glass beads and centrifuged for 5 min at 1000 g, and the supernatant was collected. A total of 20 μg of total protein extract was resolved by SDS-PAGE using 10% acrylamide gels. For immunoprecipitations, 500 μg of proteins were prepared and precleared with a 20 μg protein A-Sepharose and protein G-Sepharose bead mixture (GE Healthcare). A total of 2 μg of anti-PCNA antibody were added to the cleared extract and incubated overnight at 4 °C. The beads were washed five times with lysis buffer at a NaCl concentration of 220 mM. The resulting immunoprecipitates were loaded for SDS-PAGE using acrylamide gels.

## 3. Results

### 3.1. Increased Mutagenesis and Unequal Sister Chromatid Recombination in Pol3-01 Mutants

The defective polymerase δ of *pol3-01* mutants shows no detectable exonucleolytic activity in vitro [34,35]. *pol3-01* strains were found to have high mutagenesis levels, a fact that was attributed to its lack of exonuclease function [31,34,36]. We decided to test this hypothesis in several ways.

First, we measured the level of mutagenesis in wild-type and *pol3-01* strains using three different assays: a base substitution at the *TRP1* gene, forward mutation at the *CAN1* gene, and reversion of a 14A stretch insertion at the *LYS2* gene. The only way to revert the *trp1-289* mutation is by a very specific base substitution of the type promoted by Polζ activity. In contrast, the *CAN1* assay can detect all types of mutations [37]. Most mutations in the *LYS2* assay are either the deletion of two nucleotides or the insertion of a single adenine caused by polymerase slippage and independent of TLS polymerase activity [38]. Figure 2A shows that mutation levels were increased in the *pol3-01* mutant cells in the three assays used (19-fold, 25-fold, and 50-fold, respectively).

In addition to the mutagenic bypass of lesions by trans-lesion synthesis polymerases, an alternative DNA repair pathway uses the sister chromatid as a template to carry out an error-free bypass ([16,17]). This sister chromatid recombination cannot be directly detected, as the information in the two sister chromatids is identical. We, therefore, used an assay [32] that allows following unequal sister chromatid recombination (USCR, Appendix A). In this assay, *pol3-01* strains show a rate 2.5-fold higher than that of the wild-type control (Figure 2B).

### 3.2. Reduced Interaction between Pol3-01 and PCNA

PCNA is a central regulator of lesion bypass mechanisms. To investigate the physical interaction between Pol3-01 to PCNA (Pol30), we used the Yeast Two-Hybrid assay. Appendix A shows that Pol3-01 has a weaker interaction with PCNA than the WT Pol3 protein. These results were confirmed by co-immunoprecipitation (co-IP) experiments (Figure 2C): PCNA was immunoprecipitated, and the level of FLAG-tagged versions of Pol3 or Pol3-01 co-IPed was measured. The *pol3-01* strain exhibited a 4-fold reduction in co-IP. If, indeed, Pol3-01 interacts in a weaker manner than Pol3 with PCNA, we expect it to have less accumulation of Pol3-01 on chromatin. In Figure 2D, we show that, indeed, *pol3-01* accumulates less on the chromatin compared to WT (about a quarter of the wild-type level).

Polδ consists of three subunits; Pol3, Pol31, and Pol32 [8,9]. Pol3 interacts with Pol31, which interacts with Pol32 [13] (Figure 1A). The *pol3-01* allele carries two mutations at residues 321 and 323 (D321A, E323A). According to the recently published crystal structure of Polδ [13], these changes may disturb the interaction of the Pol3-01 protein with Pol31, and through it, with Pol32 and PCNA. It is thus possible that the weaker interaction with PCNA seen (Figure 2C) reflects the indirect interaction with Pol32 (Figure 1B).

To test whether the elevated mutation rate and the elevated sister chromatid recombination rate of *pol3-01* are due to lower attachment of the Polδ subunit to PCNA, we changed the weak PCNA-interacting peptide (PIP) motif of *pol3-01* to that of *POL32 (pol3-01-pip32)*, thus strengthening the interaction with PCNA (Figure 1C). We carried out a co-IP experiment: PCNA was immunoprecipitated, and the amount of *pol3-01* or *pol3-01-pip32* co-IPed was measured. As can be seen in Figure 3A, *pol3-01-pip32* exhibited a 4-fold increase in co-IP compared to *pol3-01*. In addition, adding the PIP motif of *POL32* to the wild type or the Pol3-01 protein led to a ~3-fold increase in their levels in the chromatin fraction (Figure 3B). Thus, indeed, Pol3 proteins bearing the PIP32 motif show increased affinity for PCNA and increased chromatin localization.

Next, we checked whether strengthening *pol3-01′*s interaction with PCNA affects the rates of mutation and sister chromatid recombination. Figure 3C shows that adding to *pol3-01* the PIP motif of Pol32 (*pol3-01-pip32*) reduces the mutation rate to about a tenth of its value. It also slightly reduces sister chromatid recombination (Figure 3C). Taken together, these results suggest that increasing the weak interaction between Polδ and PCNA, using a stronger PIP motif suppresses the high mutation and recombination rates of *pol3-01* cells.

The CysB region of Pol3 plays an important role in the interaction between Pol3 and Pol31 [13,39]. Indeed, mutations in this region, such as *pol3-11*, confer a temperature-sensitive phenotype. The *pol31-K358E* allele of Pol31 was found to be a suppressor of *pol3-11* [40,41]. The lysine 358 residue physically interacts with the Cys domains of Pol3 [39]. We, therefore, measured the level of Pol3 and Pol3-01 on chromatin in the *pol31-K358E* background. Figure 4A,B show that, indeed, the level of both versions of Pol3 is increased.

We also checked the mutation rate of *pol3-01* in a *pol31-K358E* background. The *pol31-K358E* mutation completely suppresses the increased mutagenesis rates of *pol3-01* (Figure 4C). In contrast, sister chromatid recombination was strongly elevated (Figure 4D)

To conclude so far, our results show that *pol3-01* increases both mutagenesis and sister chromatid recombination and that this increase depends on *pol3-01′*s chromatin level; if we strengthen the interaction between *pol3-01* and PCNA, we eliminate the increase in mutation rate. In contrast, the effects on USCR differed between *pol3-01-pip32* and *pol3-01 pol31-K358E*. Below, we dissect their potential mechanism of action.

The trans-lesion synthesis (TLS) pathway uses an alternative polymerase that replaces the replicative polymerases and allows the bypass of the lesion. The main TLS polymerase in yeast is Polζ, composed of Rev3 (the catalytic subunit), Rev7, Pol31, and Pol32 [39]. Since we showed that *pol3-01* increases the error-prone pathway, we asked whether this increase is *REV3*-dependent. We used the *trp1-289* mutation rate assay, which measures the rate of base substitution mutations, the main type of mutations induced by Rev3 [42].

According to Figure 5A, 75% of the increased mutagenesis of *pol3-01* is *REV3*-dependent. Thus, the lack of exonucleolytic activity in the *pol3-01* mutant only accounts for a quarter of the increased mutagenesis, and the main creator of mutations is the Polζ TLS polymerase. Since other TLS polymerases (Rev1, Rad30) may also contribute to the phenotypes, the contribution of the lack of exonucleolytic activity may be even minor.

Considering these results, our hypothesis is that *pol3-01′*s low interaction with PCNA facilitates a more efficient replacement of the replicative Polδ by Polζ, thus increasing the mutation rate in a *REV3*-dependent manner. A corollary of this model is that we expect to observe a higher level of Rev3 at chromatin in *pol3-01* strains in comparison to the WT.

As can be seen in Figure 5B, in *pol3-01′*s background, there is indeed an increased level of Rev3 on the chromatin, compared to *POL3* and *pol3-01-pip32* backgrounds.

These results strengthen our theory: *pol3-01′*s low interaction with PCNA allows the replacement of Pol3 by Rev3. This replacement allows the usage of the TLS pathway. When we strengthen the interaction between *pol3-01* and PCNA by using *PIP32*, we obtain a higher level of Pol3 on the chromatin, but a lower level of Rev3 on the chromatin, compared to *pol3-01*, in addition to a decreased mutation rate (Figure 5B).

Interestingly, the increase in recombination seen in *pol3-01* is eliminated by deleting *REV3* (Figure 5C). This result implies that the replacement of Polδ by Polζ in the presence of the *pol3-01* mutant also causes an elevation in the USCR rate.

We used a triply-tagged strain in order to measure the interactions between Pol31, Pol3, and Rev3. A co-immunoprecipitation experiment was carried out by immunoprecipitating HA-tagged Pol31 and measuring the level of co-IP by Western blot, using anti-Myc antibodies able to detect both Pol3 and Rev3. Figure 6A confirms that in *pol3-01* cells, the interaction between Pol31 and the mutant Pol3 subunit is weak and is restored to normal levels in the *pol31-K358E* background. The figure also shows that the interaction between Rev3 and Pol31 is almost undetectable in wild-type cells, but it increases in *pol3-01* and in both the *POL3* and *pol3-01* genetic backgrounds when *POL31* is mutated.

Although we demonstrated that in the *pol31-K358E* background there are higher levels of Pol3 (or Pol3-01) on the chromatin (Figure 4A,B), and indeed more interaction of the catalytic subunit with Pol31 (Figure 6A), Figure 6B shows that the single *pol31-K358E* mutant also exhibits a 4-fold increase in accumulation of Rev3 on chromatin. The increase in Rev3 at the chromatin fraction can also be seen in the *pol3-01 pol31-K358E* double mutant (Figure 6C).

In summary, our results show two very different mechanisms by which the increased mutagenesis of *pol3-01* can be suppressed: Strengthening the interactions with PCNA by mutating Pol3’s PIP reduces both mutation and sister chromatid recombination, and this result can be explained by the increase in Pol3-01 and a decrease in Rev3 at the chromatin. In contrast, the *pol31-K358E* mutation increases Pol31’s interactions with both Pol3 and Rev3. Although we see a reduction in mutagenesis, we also see an increase in the rate of sister chromatid recombination, both in the single *pol31-K358E* and in the *pol3-01 pol31-K358E* double mutant (Figure 4D).

## 4. Discussion

Timely and accurate replication of the genome is essential for life. DNA polymerases must balance the need for speedy activity with a requirement for accuracy. The proofreading mechanism of DNA polymerases, which detects their own errors while copying the genome, prevents the incorporation of mismatched nucleotides. The exonuclease activity removes the mistaken nucleotide, and a new, suitable nucleotide is incorporated instead [36,43,44]. This requires a change in the pace of progression and probably a change in the 3D architecture of the enzyme [45].

We have investigated *pol3-01*, a *POL3* mutant that lacks exonuclease function and exhibits increased rates of mutation and sister chromatid recombination. Our results show that in addition to a defect in exonucleolytic activity, Pol3-01 has reduced interaction with PCNA and a lower protein level at the chromatin (Figure 2).

To test whether the interaction with the replicative clamp plays a role in the increased levels of mutation and unequal sister chromatid recombination, we looked for different ways to strengthen the interaction between Pol3-01 and PCNA. We used two different methods:

(1) A change of the weak PCNA-interacting motif to the stronger PIP of Pol32. (2) Introducing *pol31-K358E*, a mutation in Pol31 that strengthens the interactions between Pol3 and Pol31, and thus ensures better attachment of Polδ to PCNA. Our results show that both strategies were successful in stabilizing Pol3 (Figure 3 and Figure 4) and substantially reduced the high mutation rate of *pol3-01.*

Since the exonucleolytic activity of Pol3 plays an important role in providing accuracy to the enzyme, it is logical to assume that the increased mutagenesis of *pol3-01* is a direct result of the misincorporation by the mutated DNA polymerase. However, we show (Figure 5A) that two-thirds of the base substitutions in *pol3-01* strains depend on the activity of the Polζ trans-lesion synthesis polymerase. Our results may thus explain why mutations in the exonucleolytic domain of Polδ result in a much higher rate of mutation than similar defects in the catalytic subunit of Polε [46] if only the first and not the latter facilitate the exchange with Polζ.

Since the high mutagenesis observed depends on the interactions with PCNA and on Rev3, we propose that the high level of mutations observed in *pol3-01* strains is due to the fact that the Pol3-01 protein tends to fall off chromatin, allowing an easier replacement of Polδ by Polζ (Figure 1B). Misincorporation of nucleotides by Polζ thus accounts for most of the increased mutagenesis.

In support of this hypothesis, we show that the level of Pol3-01 on chromatin is reduced (Figure 2), whereas a higher amount of Rev3 can be seen in the chromatin fraction in *pol3-01* cells (Figure 5B). Changing the PIP motif of Pol3-01 to that of the Pol32 protein increases the stability of Polδ, leading to higher protein levels in chromatin (Figure 3) and reducing the levels of Rev3 to those of the wild type (Figure 5B).

The facilitated exchange of DNA polymerases (between Pol3-01 and Polζ) may not necessitate PCNA ubiquitination. Indeed, trans-lesion synthesis in the absence of PCNA ubiquitination has been observed both in yeast and human cells [47,48].

An alternative to the error-prone lesion bypass mechanism is the error-free template switch (TS) pathway. The molecular details of this process are still being delineated; we only know that it is catalyzed by Rad5, a protein that has both helicase and poly-ubiquitination activities [16,17]. Our current understanding is that during TS, information is copied from the recently created sister chromatid in a process that results in a sister chromatid recombination. Since the two chromatids created during DNA synthesis are identical and thus indistinguishable from each other, we used unequal sister chromatid recombination (USCR) as an assay for TS. *pol3-01* strains showed increased USCR levels (Figure 2B), and these were decreased by replacing the PIP of Pol3 with a stronger PIP motif (Figure 3D). Thus, increasing Polδ affinity for PCNA via its PIP motif reduces the use of both the error-prone and the error-free branches of the DNA Damage Tolerance pathway.

We took advantage of the *pol31-K358E* mutation, which strengthens the interactions between Pol31 and Pol3 and, thus, indirectly, with Pol32 and PCNA [39]. Figure 4A,B and Figure 6A show that, similar to the PIP mutation, *pol31-K358E* leads to increased interactions with Pol3-01 and elevated levels of Polδ on the chromatin. Consistently, we also see a dramatic reduction in the level of mutations in a *pol3-01 pol31-K358E* double mutant (Figure 4C). However, in contrast to the *pol3-01-pip32* strain, the double mutant exhibits higher, not lower, levels of USCR (Figure 4D). Thus, although both the *POL3* PIP change and the *POL31* mutation have similar effects on the error-prone branch, they differ in their effect on the error-free branch of the DDT.

When we measured the level of Rev3 on the chromatin, we saw that in contrast to cells carrying the *pol3-01-pip32* allele, *pol3-01 pol31-K358E* strains exhibited higher, not lower, levels of Rev3, in addition to having higher levels of Pol3-01 (Figure 6). Thus, *pol31-K358E* strengthens the interaction of Pol31 with both Pol3 and Rev3. The reduced levels of mutation are thus probably due to a higher affinity of Pol3 over Rev3 in the binding to the mutant Pol31 subunit.

Deletion of *REV3* in the *pol3-01* strain resulted in a reduction in USCR (Figure 5C). This result uncovers an unexpected role of Polζ in the TS branch of the DDT. This role is minor in the wild type (no changes in USCR are observed in single *rev3∆* mutants) but becomes visible when the stability of Pol3 is compromised by the *pol3-01* mutation. It is possible that the instability of Pol3-01 facilitates the use of alternative repair pathways, such as the error-free DDT branch or the microhomology-mediated break-induced replication (MMBIR), which also depends on Rev3 [49].

In summary, we present evidence for the fact that the increased mutagenesis in *pol3-01* mutants is mainly due to its lower stability on the chromatin and not only due to lack of exonucleolytic activity. Since mutations in the exonuclease domain of DNA polymerases are common in various types of cancer in humans [29,30], it is particularly interesting to determine whether they also lead to increased mutagenesis that depends on TLS polymerases.

## Figures and Tables

**Figure 1 genes-14-00391-f001:**
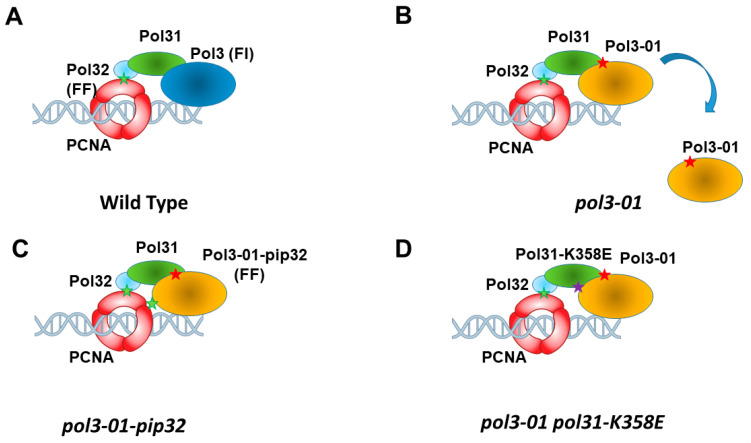
Schematic representation of Polδ in contact with PCNA. (**A**). In wt cells, Pol32 binds PCNA via its strong PIP (PCNA-interacting peptide), which contains the sequence FF (green star). Pol3 binds Pol31, which in turn binds Pol32. Pol3 sequence contains a weak PIP (FI). (**B**). In *pol3-01* cells, the Pol3 subunit carrying the mutation (red star) shows lower affinity and tends to disengage. (**C**). Changing the weak PIP of Pol3 by that of Pol32 suppresses most phenotypes of *pol3-01* cells. (**D**). The K358E mutation in *POL31* (purple star) allows higher affinity of Pol3 (and of Rev3, see text).

**Figure 2 genes-14-00391-f002:**
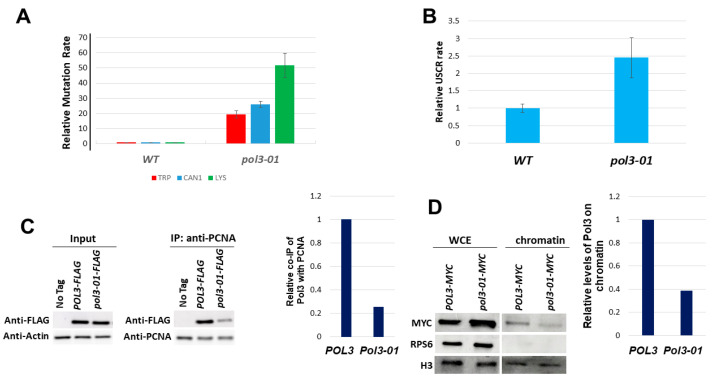
Phenotypes of *pol3-01* cells. (**A**). Relative rate of mutation of *pol3-01* cells in three different mutagenesis assays. The rate of the wild type is set to 1 in each assay. The wild-type rates for Trp+, CanR, and Lys+ mutations were 1, 2, and 1 × 10^−7^. (**B**). Relative rate of unequal sister chromatid recombination (as described in Appendix A). The rate of the wild type in this assay is 5 × 10^−5^. (**C**). co-Immunoprecipitation levels of Pol3-FLAG or Pol3-01-Flag with PCNA. (**D**). Effect of the *pol3-01* mutation on the level of Myc-tagged Pol3 protein on chromatin. Mid-log cells were fractionated into chromatin and non-chromatin fractions and probed with anti-Myc antibodies. Histone H3 served as a loading marker. WCE: whole cell extract before fractionation.

**Figure 3 genes-14-00391-f003:**
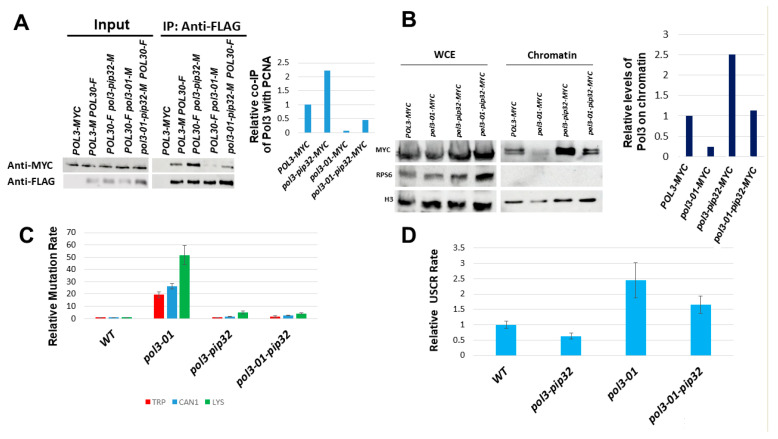
Effects of changing Pol3’s PIP by that of Pol32. (**A**). Co-IP assay between Myc-tagged Pol3/*Pol3-01/Pol3-01-pip32* and FLAG-tagged PCNA (Pol30). Proteins were immunoprecipitated with anti-FLAG antibodies. Western blotting was performed using anti-Myc to detect the presence of tagged proteins within the complexes. (**B**). Effect of the *pol3-01* mutation on the level of Myc-tagged Pol3 protein on chromatin. Mid-log cells were fractionated into chromatin and non-chromatin fractions and probed with anti-Myc antibodies. Histone H3 served as a loading marker. WCE: whole cell extract before fractionation. (**C**). Relative rate of the mutation using three different mutagenesis assays. (**D**). Relative rate of unequal sister chromatid recombination.

**Figure 4 genes-14-00391-f004:**
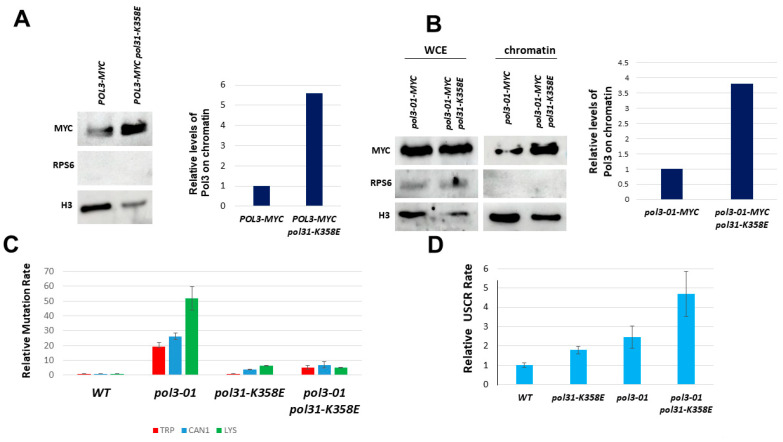
Effects of the *pol31-K358E* mutation on Pol3-01. (**A**). Fractionation analysis shows increased levels of Pol3 on the chromatin fraction. (**B**). Fractionation analysis shows increased levels of Pol3-01 on the chromatin fraction. (**C**). *pol31-K358E* suppresses the high mutagenesis rate of *pol3-01* cells. (**D**). *pol31-K358E* increases sister chromatid recombination in cells with wild-type Pol3 or with Pol3-01.

**Figure 5 genes-14-00391-f005:**
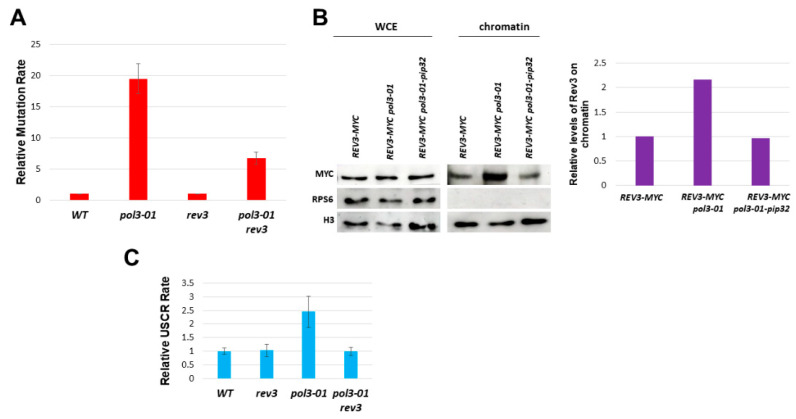
Role of Rev3 in mutagenesis and sister chromatid recombination. (**A**). The high rate of mutagenesis of *pol3-01* in the TRP assay is dependent on *REV3. (***B**). Fractionation experiment shows increased Rev3 on the chromatin fraction in *pol3-01.* The *pip32* mutation reduces its levels to those of the wild-type strain. (**C**). The increased USCR of *pol3-01* also depends on *REV3*.

**Figure 6 genes-14-00391-f006:**
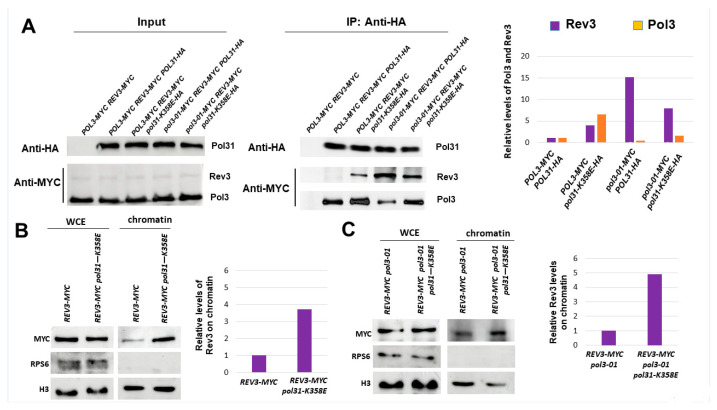
*pol31-K358E* increases the affinity of Pol3 to Rev3. (**A**). Co-Immunoprecipitation between HA-tagged Pol31 and Myc-tagged Rev3 and Pol3. Mid-log cells were immunoprecipitated with anti-HA antibodies, and Western blotting was performed using anti-Myc to detect the presence of tagged proteins within the complexes. (**B**). Fractionation analysis shows increased levels of Rev3 on the chromatin fraction in *pol31-K358E* cells. (**C**). Fractionation analysis shows increased levels of Rev3 on the chromatin fraction in *pol3-01 pol31-K358E* cells.

**Table 1 genes-14-00391-t001:** Strains used in this study.

Number	Name	Genotype
18071	E134	*MAT@ ade5-1,lys2::InsEa14,trp1-289,his7-2,leu2-3,112,ura3-52*
19779	E134 *POL3-MYC POL30-FLAG*	*MATa ade5-1,lys2::InsEa14,trp1-289,his7-2,leu2-3,112,ura3-52, POL3-MYC-TRP1 POL30-FLAG-KanMX*
19780	E134 *POL30-FLAG POL3-01-MYC*	*MATa ade5-1,lys2::InsEa14,trp1-289,his7-2,leu2-3,112,ura3-52, pol3-01-MYC-TRP1 POL30-FLAG-KanMX*
BLX1	E134 *rev3*	*MAT@ ade5-1,lys2::InsEa14,trp1-289,his7-2,leu2-3,112,ura3-52 rev3:KanMX del*
BLX2	E134 *rev3 pol3-01-V5*	*MAT@ ade5-1,lys2::InsEa14,trp1-289,his7-2,leu2-3,112,ura3-52 pol3-01-V5-KanMX rev3:KanMX*
BLX3	E134 *POL3-V5*	*MAT@ ade5-1,lys2::InsEa14,trp1-289,his7-2,leu2-3,112,ura3-52 POL3-V5-KanMX*
BLX4	E134 *pol3-01-V5*	*MAT@ ade5-1,lys2::InsEa14,trp1-289,his7-2,leu2-3,112,ura3-52 pol3-01- V5-KanMX*
BLX5	E134 *pol3-01-pip32-V5*	*MAT@ ade5-1,lys2::InsEa14,trp1-289,his7-2,leu2-3,112,ura3-52 pol3-01-pip32-V5;KanMX rev3:KanMX*
BLX6	E134 *POL3-MYC REV3-MYC pol31-HA*	*MAT@ ade5-1,lys2::InsEa14,trp1-289,his7-2,leu2-3,112,ura3-52 POL3-MYC-TRP1 REV3-MYC-HygMX pol31-HA-NatMX*
BLX7	E134 *pol3-01-MYC REV3-MYC pol31-HA*	*MAT@ ade5-1,lys2::InsEa14,trp1-289,his7-2,leu2-3,112,ura3-52 pol3-01-MYC-TRP1 REV3-MYC-HygMX pol31-HA-NatMX*
BLX8	E134 *POL3-MYC REV3-MYC pol31-K358E-HA*	*MAT@ ade5-1,lys2::InsEa14,trp1-289,his7-2,leu2-3,112,ura3-52 POL3-MYC-TRP1 REV3-MYC-HygMX pol31-K358E-HA-NatMX*
BLX9	E134 *pol3-01-MYC REV3-MYC pol31-K358E-HA*	*MAT@ ade5-1,lys2::InsEa14,trp1-289,his7-2,leu2-3,112,ura3-52 pol3-01-MYC-TRP1 REV3-MYC-HygMX pol31-K358E-HA-NatMX*
BLX10	E134 *POL3- MYC POL30-FLAG*	*MAT@ ade5-1,lys2::InsEa14,trp1-289,his7-2,leu2-3,112,ura3-52 POL3- MYC-TRP1 POL30-FLAG-KanMX*
BLX11	E134 *POL3-pip32-MYC POL30-FLAG*	*MAT@ ade5-1,lys2::InsEa14,trp1-289,his7-2,leu2-3,112,ura3-52 POL3-pip32-MYC-TRP1 POL30-FLAG-KanMX*
BLX12	E134 *pol3-01- MYC POL30-FLAG*	*MAT@ ade5-1,lys2::InsEa14,trp1-289,his7-2,leu2-3,112,ura3-52 pol3-01-MYC-TRP1 POL30-FLAG-KanMX*
BLX13	E134 *pol3-01-pip32-MYC POL30-FLAG*	*MAT@ ade5-1,lys2::InsEa14,trp1-289,his7-2,leu2-3,112,ura3-52 pol3-01-pip32-MYC-TRP1 POL30-FLAG-KanMX*
18178	PJ694	*MATa MAT trp1-901 leu2-3,112 ura3-52 his3-200 gal4del gal80del GAL2-ADE2 LYS2:: GAL1-HIS3 met2::GAL7-lacZ*
19850	BLS2	*MATa ura3-52, trp1del1, leu2-3, 112 lys2::LTR, His3::lys::ura3-, can1-101, ade2-o ade3,ade3::ura3-::ade3 in chrom. III*
18740	BLS2 *pol3-01*	*MATa ura3-52, trp1del1, leu2-3, lys2::ty::sup, His3::lys::ura3, can101, ade2-o, ade3 ade3::ura3-::ade3 in chrom. III pol3-01-myc-TRP1*
19217	BLS2 *rev3:KanMX*	*MATa ura3-52, trp1del1, leu2-3, 112 lys2::LTR, His3::lys::ura3-, can1-101, ade2-o ade3,ade3::ura3-::ade3 in chrom. III rev3:KanMX*
19257	BLS2 *rev3 pol3-01*	*MATa ura3-52, trp1del1, leu2-3, 112 lys2::LTR, His3::lys::ura3-, can1-101, ade2-o ade3,ade3::ura3-::ade3 in chrom. III rev3:KanMX pol3-01-myc-TRP1*
19398	BLS2 *pol3-01-pip32*	*MATa ura3-52, trp1del1, leu2-3, lys2::ty::sup, His3::lys::ura3, can101, ade2-o, ade3 ade3::ura3-::ade3 in chrom. III pol3-01-pip32-myc-TRP1*
19781	BLS2 *REV3-Myc*	*MATa ura3-52, trp1del1, leu2-3, 112 lys2::LTR, His3::lys::ura3-, can1-101, ade2-o ade3, Ade3::ura3-::ade3 in chrom. III, REV3-MYC-HYGMX*
19782	BLS2 *pol3-01-Myc REV3-Myc*	mk166*- MATa ura3-52, trp1del1, leu2-3, 112 lys2::LTR, His3::lys::ura3-, can1-101, ade2-o ade3, Ade3::ura3-::ade3 in chrom. III, REV3-MYC-HYGMX, pol3-01-MYC-TRP1*
20286	BLS2 *pol31-K358E*	*MAT@ ura3-52, trp1del1, leu2-3, 112 lys2::LTR, His3::lys::ura3-, can1-101, ade2-o ade3, Ade3::ura3-::ade3 in chrom. III, pol31-K358E*
20288	BL2 *REV3-MYC pol31-K358E*	*MAT@ ura3-52, trp1del1, leu2-3, 112 lys2::LTR, His3::lys::ura3-, can1-101, ade2-o ade3, plk19 Ade3::ura3-::ade3 in chrom. III, REV3-MYC-HYGMX, pol31-K358E*
20290	BL2 *REV3-MYC pol3-01*	*MAT@ ura3-52, trp1del1, leu2-3, 112 lys2::LTR, His3::lys::ura3-, can1-101, ade2-o ade3, plk19 Ade3::ura3-::ade3 in chrom. III, REV3-MYC-HYGMX pol3-01-myc-TRP1*
20292	BL2 *pol31-K358E pol3-01*	*MAT@ ura3-52, trp1del1, leu2-3, 112 lys2::LTR, His3::lys::ura3-, can1-101, ade2-o ade3, Ade3::ura3-::ade3 in chrom. III, pol31-K358E pol3-01-myc-TRP1*
20294	BL2 *REV3-MYC pol31-K358E pol3-01*	*MAT@ ura3-52, trp1del1, leu2-3, 112 lys2::LTR, His3::lys::ura3-, can1-101, ade2-o ade3, plk19 Ade3::ura3-::ade3 in chrom. III, REV3-MYC-HYGMX pol3-01-myc-TRP1 pol31-K358E*
20338	BLS2 *pol3-pip32* #110	*MAT@ ura3-52, trp1del1, leu2-3, 112 lys2::LTR, His3::lys::ura3-, can1-101, ade2-o ade3, Ade3::ura3-::ade3 in chrom. III, pol3-pip32-MYC-TRP1*
20354	BLS2 *POL3-MYC*	*MAT@ ura3-52, trp1del1, leu2-3, 112 lys2::LTR, His3::lys::ura3-, can1-101, ade2-o ade3, Ade3::ura3-::ade3 in chrom. III, POL3-MYC-TRP1*
20355	BLS2 *POL3-MYC pol31-K358E*	*MAT@ ura3-52, trp1del1, leu2-3, 112 lys2::LTR, His3::lys::ura3-, can1-101, ade2-o ade3, Ade3::ura3-::ade3 in chrom. III, POL3-MYC-TRP1 pol31-K358E-NatMX*
20362	BLS2 *pol3-01-pip32 REV3-MYC*	*MAT@ ura3-52, trp1del1, leu2-3, 112 lys2::LTR, His3::lys::ura3-, can1-101, ade2-o ade3, plk19 Ade3::ura3-::ade3 in chrom. III, REV3-MYC-HYGMX, pol3-01-pip32-MYC-TRP1*

**Table 2 genes-14-00391-t002:** Plasmids used in this study.

4112	pACT2	Y2H Vector, with a LEU2 Yeast Marker, Containing the *GAL4 AD* (Amino Acids 768–881), and an *HA* Epitope tag
1960	opb63b	pACT2 carrying *POL30*
2349	YIpAM26	*AmpR*, *pol3-01*. Received from R. Kolodner as RDK3097, made by Sugino as YIpAM26, Morrison et al. EMBO, vol. 12 No. 4, 1993
1979	pGBKT7	Y2H vector, with a KanR bacterial selection and TRP1 yeast marker, containing an N-terminal Myc tag and the *GAL4* DNA binding domain (*BD*) under *ADH* promoter
4091	PGBKT7 POL3	*POL3* cloned into pGBKT7
4093	PGBKT7 pol3-pip32	*pol3-pip32* cloned into pGBKT7
4095	PGBKT7 *pol3-01*	*pol3-01* cloned into pGBKT7

## Data Availability

Not applicable.

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
