# Peer review of "A Role for the Interactions between Polδ and PCNA Revealed by Analysis of pol3-01 Yeast Mutants"

_genes, 2023, doi:10.3390/genes14020391_

Round 1

Reviewer 1 Report

In this manuscript, Nir Heyman and co-workers found that the increased mutagenesis observed in the pol3-01 mutant can be ascribed to a weaker interaction with PCNA but not to the lack of exonucleolytic activity. They nicely showed that strengthening the interaction between Pol3-01 and PCNA by different means, reduced the high mutation rate of pol3-01. Finally, they found out that the increased mutagenic phenotype was Rev3-dependent, coming up with a model in which Pol3-01 shows lower affinity for PCNA, thus allowing an easier replacement of Pol delta by the TLS polymerase zeta. 

The experiments are well performed and mostly support the conclusions. However, the text requires a thorough revision prior to publication.

Major points:

1) How is the DNA damage sensitivity of the pol3-01 mutant? Is it more resistant to DNA damage due to higher TLS activity? In the methods there is a section called “DNA damage sensitivity” but I have not found any spot assay in the current version of the manuscript...

2) I feel that some messages should be discussed in more detail. How do the authors explain an increase in template switch pathway when TLS polymerase zeta replaces pol delta more efficiently? The authors did not discuss about the relevance of PCNA-Ub on TLS. In the absence of DNA damage, PCNA should remain unmodified in yeast. Then, do the authors claim that the ubiquitylation state of PCNA is dispensable for the recruitment of Rev3 in the pol3-01 mutant? At least in mammalian cells, TLS can occur independently of PCNA-Ub (10.1371/journal.pgen.1002262), but, is it possible in yeast?

Minor points:

1) It is a bit confusing that in the text the authors explain the three assays to measure mutation rate in an order that is different to the shown in Figure 2A. It will help to change the order in the graph (1st CAN1, 2nd LYS2, 3rd TRP).

2) The text in some figures is too small (per example Figure 2C)

3) Figure S2. What it is 3-AT and what it is used for is not explained neither in the legend nor in the method section.

4) Line 126, correct Figure 2A for Figure 1A.

5) It is not explained in the methods how the PIP motif of pol3-01 was changed to the one from POL32. Site-directed mutagenesis?

6) Some parts of the text seem to be written in a different font, per example, see lines 134 and 135. Please, correct it.

7) Figure 3A, the legend says “Actin was used as a loading control”. Where is the actin blot?

8) Figure 4A. WCE should be shown.

9) Figure 4B is not cited in the text.

10) Figure 6A, the title in the y axis of the quantification is wrong (it says “on chromatin”). Furthermore, the quantification of Pol3 in the pol3-01-myc pol31-K358E-HA strain does not seem accurate: by eye Pol3 levels are similar to the ones in the POL3-myc POL31-HA strain, but the quantification shows that there is a 5-fold increase.

11) Sentence in line 211 should be rephrased: according to the co-IP, pol31-K358E leads to stronger interaction of the protein with Pol3 and Rev3 but only in the wild type Pol3. In the case of the Pol3-01 version of the protein, there is a stronger interaction of the protein with Pol3 but a weaker interaction with Rev3.

12) Figure 6B and 6C legends are missing. 

13) Line 220-221, it reads: “in contrast to the effect strengthening the catalytic subunit’s PIP, which only showed an effect in the pol3-01 background (Figure 3B)” 

Why is it claimed here that only showed an effect in the pol3-01 mutant while in the line 137, it is claimed that the effect is in both wild type and Pol3-01 proteins? In fact, figure 3B shows an effect in both.

14) Line 266, Cite Figure 5B instead of Figure 3C.

15) Line 274 Figure X?

16) Lines 184-186. The putative role of other TLS polymerases, such as Rev1, in the increased mutagenesis is not formally excluded. Thus, it is not sufficiently proven that “the remaining 25% of the mutagenesis is due to the lack of exonucleolytic activity of pol3-10”. Under my opinion, the sentence should be tone down.

Reviewer 2 Report

Faithful DNA replication is essential to maintain genome integrity. In S phase, with help from PCNA, DNA polymerases coordinate to achieve high fidelity DNA synthesis avoiding mutagenesis. According to previous literature, pol3-01, with mutations in exonuclease domain responsible for proofreading function, were the reason contributing to the high level of mutagenesis and template switch. In this manuscript, the authors carefully reviewed the causes and effects with several genetics and cellular biological experiments, and argued that the reason is because of the weaker interaction between pol3-01 and PCNA, rather than the exonuclease activity of pol3-01. However, the genetic outcomes need further evidence to support the statement before publishing.

1.       In pol3-01 strain, the authors showed higher mutation rate of three different reporter genes, which relies on Pol ζ. Mechanistically, the authors proposed a polymerase switch from Pol δ to Pol ζ. Is this polymerase switch occurring upon PCNA ubiquitination by activating DDT?

2.       Can the authors use the mutation signature of pol3-01 and other strains to better support the idea the increased mutation rates is caused by a Pol ζ engagement rather than the exonuclease mediated proofreading.

3.       It was interesting that REV3 is required for both mutagenesis and USCR rate in pol3-01 strain. Is it possible that the USCR is utilizing a MMBIR mechanism (Cynthia J. Sakofsky, et al. Molecular Cell 2015)?

4.       The co-IP and chromatin fractionation experiments should be reproduced for quantification.

5.       The authors may also need to confirm the exonuclease activity of the key strains in this work pol3-01-pip32 and pol31-K358E. 
